# Long-Term Immunogenicity of Inactivated and Oral Polio Vaccines: An Italian Retrospective Cohort Study

**DOI:** 10.3390/vaccines10081329

**Published:** 2022-08-17

**Authors:** Angela Maria Vittoria Larocca, Francesco Paolo Bianchi, Anna Bozzi, Silvio Tafuri, Pasquale Stefanizzi, Cinzia Annatea Germinario

**Affiliations:** 1Hygiene Department, Bari Policlinico General Hospital, 70124 Bari, Italy; 2Interdisciplinary Department of Medicine, Aldo Moro University of Bari, Piazza Giulio Cesare 11, 70124 Bari, Italy

**Keywords:** eradication, healthcare workers, poliomyelitis

## Abstract

Oral and inactivated poliovirus (PV) vaccines have contributed toward the global eradication of wild PV2 and PV3, as well as the elimination of PV1 in most countries. While the long-term (>5–10 years) persistence of protective antibodies in ≥80% of the population vaccinated with ≥3–4 doses of oral poliovirus vaccine (OPV) has been demonstrated, the duration of immunity in people vaccinated with the inactivated poliovirus vaccine (IPV) is still unclear. This study evaluated the seroprevalence of anti-PV neutralizing antibodies and the long-term immunogenicity conferred by OPV and IPV in a sample of medical students from the University of Bari (April 2014–October 2020). The levels of neutralizing PV1, PV2, and PV3 antibodies in blood samples taken during the assessments were evaluated. Neutralizing antibodies against PV1, PV2, and PV3 were present in >90% of the study participants, with rates of >99%, >98%, and ~92–99%, respectively. IPV resulted in a higher immunological response than OPV against PV3. Protective antibodies against all three viruses persisted for at least 18 years after administration of the last vaccine dose. Until PV1 is completely eradicated, maximum vigilance from public health institutions must be maintained.

## 1. Introduction

The eradication of polioviruses remains a major global public health goal. The introduction of the inactivated poliovirus vaccine (IPV) and trivalent oral poliovirus vaccine (OPV) in official vaccination schedules worldwide has led to the eradication of wild PV2 (in 2015) and wild PV3 (in 2019); moreover, since 2017, wild PV1 cases have only been reported in Afghanistan and Pakistan [1,2,3,4]. Nonetheless, the WHO’s strategy to eradicate polio might slow down in situations of conflict (i.e., in which socioenvironmental and hygienic conditions are disrupted) [5].

In 1964, the Italian Ministry of Health developed a mass vaccination campaign in which the Sabin vaccine was offered free and actively to all children between the ages of 6 months and 14 years [6]. Between 1964 and 2000, vaccinations with OPVs resulted in a small number of cases of vaccine-associated paralytic poliomyelitis. Due to ethical concerns and the favorable epidemiological context, in 2000, a sequential schedule (IPV–IPV–OPV–OPV) was introduced. In 2003, the use of a live attenuated vaccine was suspended and IPV was introduced exclusively for polio vaccinations during childhood [6]. Since 2002, the vaccination schedule in Italy has consisted of the first three doses of IPV to infants at 3, 5, and 11 months of age using a hexavalent formulation (IPV–hepatitis B–Haemophilus influenzae type b–tetanus–diphtheria–acellular pertussis), with a fourth dose administered as a tetravalent formula (tetanus–diphtheria–acellular pertussis–IPV) at 5–6 years of age. In 2017, a fifth dose administered during adolescence was recommended. Moreover, in 2017, the Italian government made vaccinations against polio mandatory for infants and children [7]. With the success of vaccination campaigns carried out since 1964, Italy (together with the entire European region) was certified as polio-free in 2002 by the Regional Commission for the Certification of Poliomyelitis Eradication; in fact, no case of polio had been recorded since 1983 [8].

Serologic studies have shown that seroconversion rates—following three doses of either IPV or OPV—are nearly 100% for all three viruses [9]. However, while the World Health Organization (WHO) reported strong scientific evidence for the long-term (>5–10 years) persistence of protective antibodies in ≥80% of the population vaccinated with ≥3–4 doses of OPV [10], the duration of immunity conferred by IPV is unclear [11].

The aim of this study was to evaluate the seroprevalence of anti-poliovirus neutralizing antibodies in a sample of medical students and residents from the medical school of the University of Bari who had been fully vaccinated with the IPV. The long-term immunity of participants who received OPV was also determined and compared with that of the IPV group. The study was carried out in Apulia (southern Italy, with ~4,000,000 inhabitants).

## 2. Materials and Methods

This was a retrospective cohort study.

The study population was composed of students and residents who attended the Hygiene Department from April 2014 to October 2020. Inclusion criteria were: vaccinated with four doses of IPV or all OPV, according to the Italian schedule that was in effect until 2017 (3 doses during the first year of life and the fourth dose at age 5–6 years). Individuals without available vaccination histories, who were never vaccinated, who lived for more than a year in a highly endemic country, vaccinated with both the IPV and OPV, were vaccinated with another formula, or who had been vaccinated with less or more than four doses of IPV or OPV at baseline were excluded from the study. None of the study candidates reported a history of poliomyelitis.

From April 2014 to October 2020, 6105 medical students and residents were tested; a blood sample was taken during the first access to the clinic. The immunization status, downloaded from Apulia’s Regional Immunization Database (GIAVA), was available for 4661/6105 (76.3%). From this group, 123/4661 (2.6%) had received four doses of IPV and were included in the study; the other subjects were vaccinated as follows: 1408 (30.2%) vaccinated with four doses of trivalent OPV, 945 (20.3%) received a mixed schedule (IPV–OPV), 2036 (43.7%) received less or more than four doses of trivalent OPV and 149 (3.2%) with less or more than four doses of IPV. Those included participants were matched with a control group consisting of individuals who attended the same biological screening program and had been vaccinated with four doses of trivalent OPV. An allocation ratio of 1:3 was used to improve the statistical analysis power. The two groups were matched for age and sex using STATA MP16 software, resulting in a final sample of 492 individuals: 123 who had been vaccinated with four doses of IPV and 369 with four doses of trivalent OPV.

### 2.1. Laboratory Analysis

The neutralization test was conducted in microtiter plates according to the guidelines of the WHO/Expanded Program on Immunization (EPI). Titers ≥ 1/8 were considered positive, as recommended by the WHO/EPI [12]. Two-fold dilutions of inactivated sera (from 1/8 to 1/1024) were incubated in duplicate with suspensions of each of the three reference Sabin strains (PV1/Mahoney strain, PV2/MEF-1 strain, and PV3/Saukett strain) corresponding to a 100 TCID50/0.025-mL challenge. After a 3-h incubation at 36 °C, 5% CO_2_, a human heteroploid Hep-2 cell suspension (1–2 × 104 cells/0.1 mL; MEM Earle’s salts 10% FBS; 37 °C, 5% CO_2_) was added to each well containing the virus–serum mixtures. A titration of each viral strain and cell controls were included. The plates were incubated at 36 °C for 5 days and then examined for the appearance of cytopathic effects (CPE) using an inverted microscope. The neutralizing antibody titer (expressed as reciprocal) was determined using the Karber formula, based on the highest dilution of serum that protected 50% of the cultures against a 100 TCID50 viral challenge and inhibited CPE. Titers ≥ 1/8 were considered positive, as recommended by the WHO/EPI.

### 2.2. Statistical Analysis

The data were analyzed using STATA MP16 software. Continuous variables were reported as the mean ± standard deviation and range, and categorical variables as proportions, with 95% confidence intervals (95%CIs) when appropriate. Protective antibody titers were classified as low (1/8–1/32) or high (1/64–>1/256) and compared by group (IPV vs. OPV) and age class. Skewness and kurtosis tests were used to evaluate the normality of the continuous variables, but none of them were normally distributed or normalizable. Wilcoxon’s rank sum test was used to compare continuous variables between groups and chi-squared or Fisher’s exact tests to compare proportions with respect to group and age class. To assess the seroprotection determinants at the time of enrollment (seroconversion after the vaccine basal cycle, which is three doses during the first year of life and the fourth dose at age 5–6 years), multivariate logistic regression models were created for each type of poliovirus, in which the seroprotection determinants were the outcome and group (IPV vs. OPV), sex (male vs. female), age at enrollment (years), and immune-related chronic disease (yes/no). Adjusted odds ratios (aORs) were calculated together with their 95%CIs. Protective antibody survival (PAS), defined as the time elapsed from the last dose of the routine vaccine to the evaluation of the antibody titer (years), was determined and then analyzed using Kaplan–Meier curves. The log-rank test was used to evaluate differences between groups. The loss of seroprotection per 1000 person-years and the 95%CIs were calculated. The incidence rate ratio (IRR), in which the value for the OPV group was the denominator and that for the IPV group the numerator, was also calculated together with the 95%CIs. For all tests, a two-sided *p*-value < 0.05 was considered statistically significant.

The study was carried out in accordance with the Declaration of Helsinki. All healthcare workers (HCWs) who were screened provided written consent regarding the use and scientific publication of data collected for clinical purposes.

## 3. Results

The study population included 492 subjects, of which, 472 were students (95.9%; mean age: 21.1 ± 2.6 years) and 20 were residents (4.1%; mean age: 29.1 ± 1.9 years). A total of 344 (69.9%) subjects were female; there was no significant difference between the OPV group (n = 258/369; 69.9%) and the IPV group (86/123; 69.9%; *p* = 1.000). The average age at study enrollment was 21.4 ± 3.1 years (range = 18.0–33.0), with no difference between the groups (OPV: 21.5 ± 3.0; range = 18–33 vs. IPV: 21.2 ± 3.2; range = 18–33; *p* = 0.126). The average PAS time was 19.0 ± 3.1 years (range = 9–31), specifically 19.1 ± 3.0 (range = 12–30) for the OPV group and 18.6 ± 3.5 (range = 9–31) for the IPV group.

### 3.1. PV1

The prevalence in the study population of the absence of PV1 neutralizing antibodies was 0.20% (95%CI: 0.01–1.12; n = 1/492); the difference between the OPV and IPV groups was not significant (*p* = 1.000; Table 1). A high titer was measured in 91.5% (n = 449/491) of the study participants, with no significant difference between the two groups (OPV vs. IPV: *p* > 0.05 for each PV; Table 1, Figure 1).

In the OPV group, the titer of neutralizing antibodies decreased significantly with increasing age (*p* = 0.027), whereas in the IPV group, the titer of neutralizing antibodies was slightly lower but remained relatively constant among age classes (*p* = 0.782; Figure 2).

In the multivariate logistic regression, there was no association between the seroprevalence of anti-PV1 antibodies and any of the analyzed determinants (*p* > 0.05; not shown).

The incidence of seronegativity in the whole sample per 1000 person-years was 0.10 (95%CI: 0.01–0.74). The incidence of seronegativity in the OPV group was 0.14 (95%CI: 0.01–0.98), but due to the small number of events in the IPV group, neither seronegativity nor the IRR could be calculated. There was no significant vaccine-based difference in the PAS (log-rank *p*-value = 0.594; Figure 3).

### 3.2. PV2

The prevalence in the study population of the absence of PV2 neutralizing antibodies was 1.45% (95%CI: 0.47–3.35; n = 5/345), with no significant difference between the OPV and IPV groups (*p* = 1.000; Table 1). A high titer was detected in 72.9% (n = 248/340) of the study population, with no significant difference between the groups (*p* > 0.05; Table 1). In the OPV group, the titer of neutralizing antibodies decreased significantly with age (*p* = 0.002); in the IPV group, the titer was slightly higher but also decreased with age, albeit not significantly (*p* = 0.186; Figure 3).

In the multivariate logistic regression, there was no association between the seroprevalence of anti-PV2 antibodies and the analyzed determinants (*p* > 0.05; not shown).

The incidence of seronegativity per 1000 person-years was 0.79 (95%CI: 0.32–1.85) and was lower in the IPV group (0.59; 95%CI: 0.01–4.18) than in the OPV group (0.83; 95%CI: 0.31–2.22), with an IRR of 0.71 (95%CI: 0.01–7.15; *p* = 0.830). The PAS did not differ as a function of the group (log-rank *p*-value = 0.974; Figure 3).

### 3.3. PV3

The prevalence in the study population of the absence of PV3 neutralizing antibodies was 6.50% (95%CI: 4.49–9.06; n = 32/492), with a statistically significant difference between the OPV and IPV groups (92.1% vs. 97.6%; *p* = 0.035; Table 1). A high titer was detected in 56.3% (n = 259/460) of the study population, without a difference between groups (*p* > 0.05; Table 1). The titer of neutralizing antibodies decreased significantly with age in the IPV group (*p* = 0.027) but, although similar, largely remained constant in the OPV group (*p* = 0.185; Figure 2).

In the multivariate logistic regression, an association at the limit of statistical significance was determined between the seroprevalence of anti-PV3 antibodies and the group (aOR = 3.34; 95%CI: 1.00–11.20; *p* = 0.050). There were no significant associations between any of the other analyzed determinants (*p* > 0.05; Table 2).

The incidence of seronegativity per 1000 person-years was 3.34 (95%CI: 2.36–4.72) and was lower in the IPV group (0.13; 95%CI: 0.41–3.97) than in the OPV group (4.00; 95%CI: 2.78–5.76), with an IRR of 0.32 (95%CI: 0.06–1.03; *p* = 0.037). The PAS did not differ as a function of the group (log-rank *p*-value = 0.059; Figure 3).

## 4. Discussion

Our study showed that neutralizing antibodies against all three types of poliovirus were present in >90% of the study participants, regardless of their vaccination with IPV or OPV, with rates of >99% for PV1, >98% for PV2, and ~92–99% for PV3. A higher immunological response to PV3 was obtained with IPV than with OPV (98% vs. 92%), as was also determined in the logistic and semiparametric Cox regression models. Tafuri et al., in a 2008 Italian study [13], determined seropositivity rates of >99% for all three viruses in a group of Apulian children (vaccination status unknown) and adolescents (the data are similar to data reported in studies set up in other countries) [14,15,16].

Over time, both vaccines seem to trigger an immune response that leads to high levels of neutralizing antibodies for PV1 (87–94%), lower levels for PV2 (62–85%), and even lower levels for PV3 (46–60%). The levels of neutralizing antibodies decreased with increasing age but without substantial differences between the OPV and IPV groups. This decline is a proxy for the real risk factor, which is the time elapsed since the last vaccine dose. Similar to other vaccines [17,18,19,20], the role of age (or time elapsed since the last dose) in the response to polio vaccines has been demonstrated in several studies [13,21,22,23].

The PAS analysis showed that protective antibodies against all three viruses persist for at least 18 years after the administration of the last dose of OPV or IPV; a longer duration of immunity against PV3 was provided by IPV than by OPV. Although the long duration of OPV immunization is well established [9], to our knowledge, ours is the first study to quantitatively evaluate a large study population vaccinated with four doses of the oral vaccine during childhood and to compare the two vaccine formulations that have long been in use. Our findings should be considered in light of the absence of natural boosters in Italy, where, in the last 30 years, no case of polio has been reported (or use of supplementary immunization activities (SIAs)). In addition, in the Apulia region, analyses of blood and stool samples from emigrants arriving mostly from the Middle East and Africa have likewise been negative for poliovirus [24,25].

In summary, the time between the last vaccination and the antibody titer evaluation is a determinant of the levels of persisting neutralizing antibodies. While the antibody titer decreases over time, immunity against PV1 and PV2 can possibly be considered life-long; on the other hand, a challenge dose of IPV or trivalent OPV may strengthen the long-term persistence of protective immunity, especially against PV3. There were no significant differences between IPV and OPV, although IPV may provide a higher immunological response against PV3.

The strengths of our study are in its evaluation of the long-term immunogenicity of IPV vs. OPV and the comparisons of antibody titers over time. Moreover, to our knowledge, this is the first study that compared the two formulas and one of the most important experiences in the literature regarding subjects vaccinated with IPV. Our data showed the overall higher effectiveness of the IPV formula considering the duration of immunity and prevalence of neutralizing antibodies; nevertheless, the OPV formula remains crucial in the prevention of the transmission and, therefore, it is a valid option in countries where the virus circulations are still highly probable. Nonetheless, a major limitation involved the age distribution of the study participants, which was mostly <25 years old; indeed, only 53 subjects were >25 years old (but this was expected, as our population consisted of students in medical school). This may have distorted the results since young adults have enhanced durable immune memories. Furthermore, the investigation of rare events, such as the absence of neutralizing antibodies, especially among people vaccinated with IPVs, requires studies with larger numbers of participants. Moreover, the neutralization antibody titer measurement does not value the vulnerabilities of the subjects to mucosal intestinal infections with PV and subsequent transmissions; indeed, adequate humoral immunogenicity assessments are relevant to protect against paralysis, but not against intestinal replication and transmission of poliovirus. Future studies should expand the sample size and the observation time to evaluate critical issues that may place an individual or population at risk in the event of wild virus reintroduction.

In conclusion, the basal vaccination scheme for IPV induces long-lasting protection against paralytic poliomyelitis. Wild PV2 and PV3 have been eradicated, and protection against paralysis from polio against PV1 remains close to 100% even after many years. Considering the efficacy of four doses, a fifth booster IPV dose, as recommended by the Italian immunization plan, will likely be sufficient to ensure life-long protection. As pointed out by Lopalco PL in a 2016 study [26], the global use of OPV has led to the eradication of wild PV2 and PV3, but the burden caused by vaccine-derived cases of polio is becoming increasingly problematic. These data support the use of IPVs to maintain high levels of seropositivity, particularly to PV3, accompanied by high-level clinical and environmental surveillance. Indeed, in Italy, there is active surveillance for cases of acute flaccid paralysis [27] and a high level of IPV coverage is part of the most recent Italian immunization plan [28].

## Figures and Tables

**Figure 1 vaccines-10-01329-f001:**
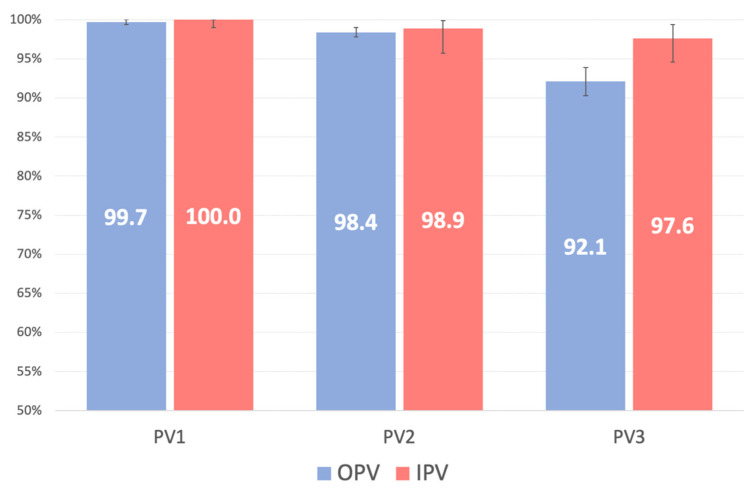
Prevalence (%) among the study participants of neutralizing antibodies, per poliovirus (PV) type.

**Figure 2 vaccines-10-01329-f002:**
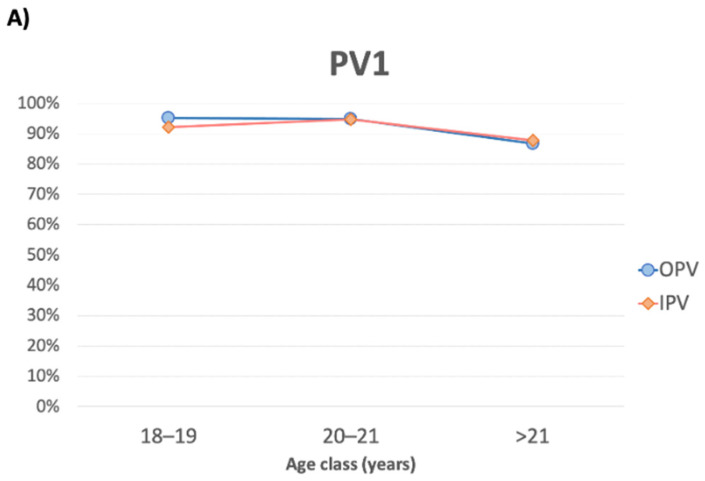
Prevalence (%) among the study participants of high protective titer of neutralizing antibodies against PV1 (**A**), PV2 (**B**), and PV3 (**C**), per age class.

**Figure 3 vaccines-10-01329-f003:**
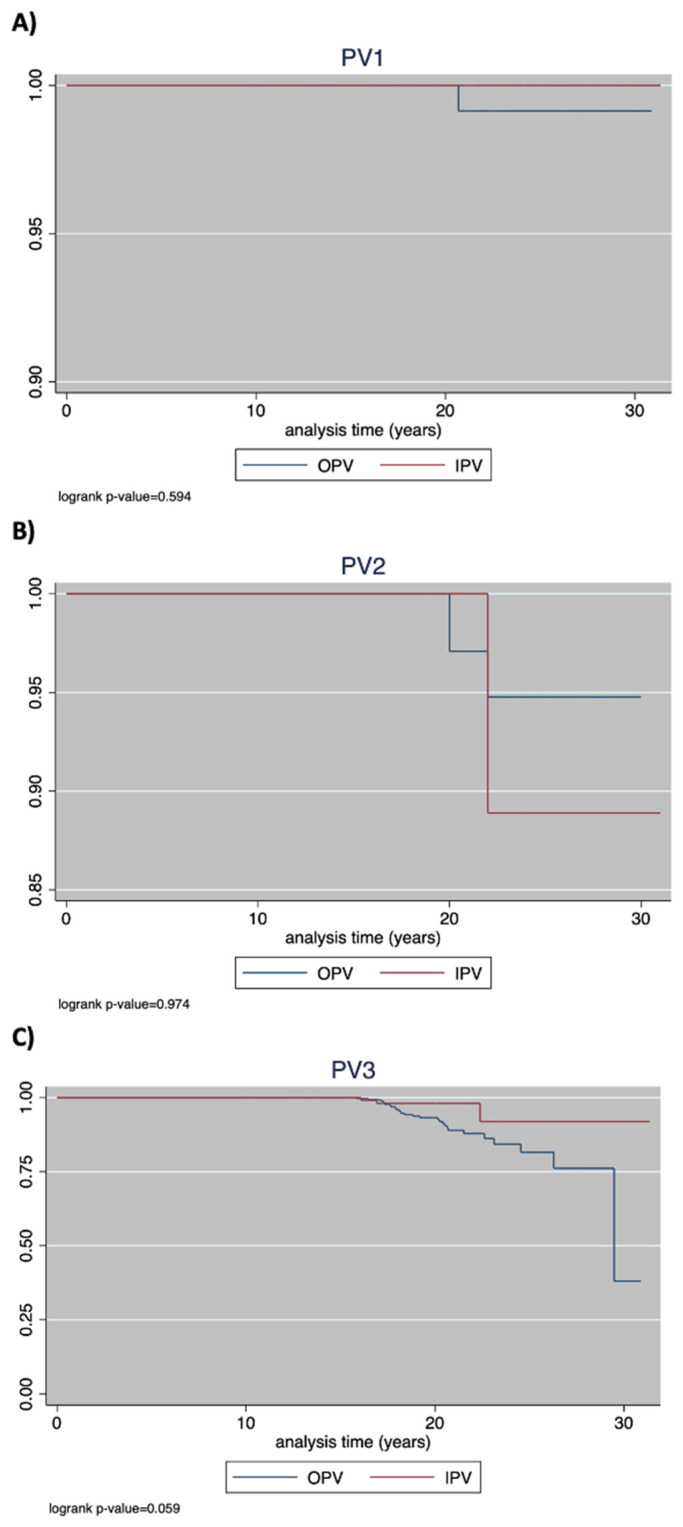
Kaplan–Meier estimates of protective antibody survival, per group (IPV vs. OPV) for (**A**) PV1, (**B**) PV2, and (**C**) PV3.

**Table 1 vaccines-10-01329-t001:** Proportion of study participants without poliovirus (PV) neutralizing antibodies and the distribution of the titer (low–high) between groups with respect to vaccination and PV type.

Variable	PV1				PV2				PV3			
	OPV	IPV	Total	*p*-Value	OPV	IPV	Total	*p*-Value	OPV	IPV	Total	*p*-Value
Susceptible; n (%; 95%CI)	1 (0.27; 0.00–1.50)	0 (0.00; 0.00–2.95)	1 (0.20; 0.01–1.12)	1.000	4 (1.59; 0.43–4.01)	1 (1.08; 0.03–5.85)	5 (1.45; 0.47–3.35)	1.000	29 (7.85; 5.33–11.09)	3 (2.44; 0.51–6.96)	32 (6.50; 4.49–9.06)	0.022
Protective titer; n (%)				0.859				0.179				0.328
low	31/368 (8.4)	11/123 (8.9)	42/491 (8.6)		72/248 (29.0)	20/92 (21.7)	92/340 (27.1)		144/340 (42.4)	57/120 (47.5)	201/460 (43.7)	
high	337/368 (91.6)	112/123 (91.1)	449/491 (91.4)		176/248 (71.0)	72/92 (78.3)	248/340 (72.9)		196/340 (57.6)	63/120 (52.5)	259/460 (56.3)	

**Table 2 vaccines-10-01329-t002:** Analysis of the determinants of neutralizing anti-PV3 antibodies in a multivariate logistic regression model.

Determinant	aOR	95%CI	*p*-Value
Group (IPV vs. OPV)	3.34	1.00–11.20	0.050
Sex (male vs. female)	0.97	0.44–2.12	0.934
Age (years)	1.00	0.89–1.13	0.979
Immune-related chronic disease (YES/NO)	1.83	0.80–4.18	0.152

aOR: adjusted odds ratio; Hosmer–Lemeshow Χ^2^ = 11.8; *p* = 0.162.

## Data Availability

Data available on request due to restrictions e.g., privacy or ethical.

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
