# Peer review of "Long-Term Immunogenicity of Inactivated and Oral Polio Vaccines: An Italian Retrospective Cohort Study"

_vaccines, 2022, doi:10.3390/vaccines10081329_

Round 1

Reviewer 1 Report

GENERAL: This study adds to the literature and provide useful data on antibody durability, particularly important in this stage of polio eradication efforts. A major limitation to this study however is the narrow age range. The authors draw attention to this somewhat in the discussion, but it should be further acknowledged that the range (18-33 years with most being <25) does not provide adequate insight into true age-related decline as this entire population is generally considering as "young adults". Further explanation of why the major groups do not expand >25 group would be helpful (how many subjects were in this older group?).

The authors often use the word "better" to describe the immune responses observed. Due to the subjective nature of this word and the fact, as the authors point out, that various aspects of vaccine-induced immunity are necessary (mucosal and systemic), I favor more specific terminology as related to the actual observations - duration of neutralizing antibody responses or sustained rates of seropositivity (as this is really a population level assessment). Future articulation of these differences in vaccine-induced immunity would strengthen the discussion. 

I do think that the higher rates or seronegativity in the OPV group are important and very worthy of highlighting as a major finding. I would shift focus here in discussion. 

SPECIFIC:
Lines 63 and 111- consider not making a separate paragraph and integrating with prior/following paragraph respectively.

Lines 140-142 - please clarify where these p values are coming from in Figure 1 and consider adding marker or significance to figure 1 where appropriate. If not, further specify what is being compared here in text. 

Line 146 - please provide the PAS time by group w ranges as well. Consider linking to Figure 2.

Table 1. Please clarify what is being compared (p value) in low vs. high in 2 study groups. Is it the ratio high/low within or between groups?

Line 157 - specify that there was higher proportion of absence of PV3 neuts in OPV group.

Line 171 - also split PAS between groups as above.

Line 175 - this statement re hospitalizations seems out of place and out of scope of this report. I actually don't understand how it fits with this study population considering all are vaccinated??

Line 180 - consider changing word "better" to "higher population level of seroprotection" or "seroprotection rate"

line 184 - type: at to as. consider rewording for clarity.

line 194 - consider changing "a longer immunity" to "longer duration of immunity again PV3..."

line 207 - type: o to on / strength to strengthen / add PV3 to the end of the sentence. 

line 209 - once again, avoid "better". substitute with "enhanced durability" or something to that effect. 

Line 231 - became => "has become" and avoid referring to the author in 3rd person. Consider; "This data supports the use of IPV to maintain high levels of seropositivity, particularly to PV3."

Author Response

Q1. This study adds to the literature and provide useful data on antibody durability, particularly important in this stage of polio eradication efforts. A major limitation to this study however is the narrow age range. The authors draw attention to this somewhat in the discussion, but it should be further acknowledged that the range (18-33 years with most being <25) does not provide adequate insight into true age-related decline as this entire population is generally considering as "young adults". Further explanation of why the major groups do not expand >25 group would be helpful (how many subjects were in this older group?).

A1. We revised discussion paragraph.

Q2. The authors often use the word "better" to describe the immune responses observed. Due to the subjective nature of this word and the fact, as the authors point out, that various aspects of vaccine-induced immunity are necessary (mucosal and systemic), I favor more specific terminology as related to the actual observations - duration of neutralizing antibody responses or sustained rates of seropositivity (as this is really a population level assessment). Future articulation of these differences in vaccine-induced immunity would strengthen the discussion.

A2. Revised.

Q3. I do think that the higher rates or seronegativity in the OPV group are important and very worthy of highlighting as a major finding. I would shift focus here in discussion.

A3. Revised.

Q4. Lines 63 and 111- consider not making a separate paragraph and integrating with prior/following paragraph respectively.

A4. Revised.

Q5. Lines 140-142 - please clarify where these p values are coming from in Figure 1 and consider adding marker or significance to figure 1 where appropriate. If not, further specify what is being compared here in text.

A5. We revised the text.

Q6. Line 146 – please provide the PAS time by group w ranges as well. Consider linking to Figure 2.

A6. Revised.

Q7. Table 1. Please clarify what is being compared (p value) in low vs. high in 2 study groups. Is it the ratio high/low within or between groups?

A7. We clarified this point in the title of the table.

Q8. Line 157 - specify that there was higher proportion of absence of PV3 neuts in OPV group.

A8. Revised.

Q9. Line 171 - also split PAS between groups as above.

A9. Revised.

Q10. Line 175 - this statement re hospitalizations seems out of place and out of scope of this report. I actually don't understand how it fits with this study population considering all are vaccinated??

A10. It was a mistake, we cancelled it.

Q11. Line 180 - consider changing word "better" to "higher population level of seroprotection" or "seroprotection rate"

A11. Revised.

Q12. line 184 - type: at to as. consider rewording for clarity.

A12. Revised.

Q13. line 194 - consider changing "a longer immunity" to "longer duration of immunity again PV3..."

A13. Revised.

Q14. line 207 - type: o to on / strength to strengthen / add PV3 to the end of the sentence. 

A14. Revised.

Q15. line 209 - once again, avoid "better". substitute with "enhanced durability" or something to that effect.

A15. Revised.

Q16. Line 231 - became => "has become" and avoid referring to the author in 3rd person. Consider; "This data supports the use of IPV to maintain high levels of seropositivity, particularly to PV3."

A16. Revised.

Reviewer 2 Report

The paper describes a simple and small population size, yet effective study of IPV versus OPV vacination effect on the presence and persistence of High titler neuralizing antibodies. Despite the initial large size of the population, after aplication of exclusion criteria only 123 IPVvaccinated  subjects reamined in the study, alongside with a larger OPV vaccinated population (1:3 IPV/OPV ratio to increase statistical power).

The study is well designed and well controled, althought limitations are still present that are properly discussed. However, some major and a few minor issues should be resolved before publication can be recomended:

Major Issues:

- line 102 refers to chonic disease, whithout specifying if irrelevant any chronic diseases were considered or if immune related chronic diseases

- letterings in graphics are far too small. In general no scale or legend is readable without substantial enlarging

- Althouth results for PV2 are showed in graphics and tables, only results for PV1 and PV3 are described in text.

- Lines 175-176 should be in matherials and methods as they refer to population characterization rather than PV3 results. Also, resons for hospitalizations should be included as they are predicably not related to Polio infection.

Minor Issues:

- use of comma as thousands separator is confusing in text where commas may appear close to the number for sentence purposes (see for example line 73)

- abreviations should always be described on first use, even if broadly used. For example HCW (presumably Health Care Workers) is used 3 times but never defined. 

- line 94 "but any of them" should be "but all of them"

- line 166 refers to table 3 but should be table 2. Also title of table 2 is labeled table 1 (line 167)

- line 185 "similar at the ones" should be "similar to the ones"

- line 207 "IPV o trivalent" should be "IPV or trivalent"

- line 228 "more than sufficient" should be changed to "sufficient"

Author Response

Q1. line 102 refers to chonic disease, whithout specifying if irrelevant any chronic diseases were considered or if immune related chronic diseases A1. We revised those points in the text.

A1. Revised.

Q2. letterings in graphics are far too small. In general no scale or legend is readable without substantial enlarging

A2. Unfortunately, we are not able anymore to modify the figure. We hope that after copy editing the editors will publish the figure bigger.

Q3. Althouth results for PV2 are showed in graphics and tables, only results for PV1 and PV3 are described in text.

A3. Revised.

Q4. Lines 175-176 should be in matherials and methods as they refer to population characterization rather than PV3 results. Also, resons for hospitalizations should be included as they are predicably not related to Polio infection.

A4. Revised.

Q5. use of comma as thousands separator is confusing in text where commas may appear close to the number for sentence purposes (see for example line 73)

A5. Revised when possible.

Q6. - abreviations should always be described on first use, even if broadly used. For example HCW (presumably Health Care Workers) is used 3 times but never defined.

A6. Revised.

Q7. - line 94 "but any of them" should be "but all of them"

A7. It is correct “none of them”.

Q8. - line 166 refers to table 3 but should be table 2. Also title of table 2 is labeled table 1 (line 167)

A8. Revised.

Q9. - line 185 "similar at the ones" should be "similar to the ones"

A9. Revised.

Q10. - line 207 "IPV o trivalent" should be "IPV or trivalent"

A10. Revised.

Q11. - line 228 "more than sufficient" should be changed to "sufficient"

A11. Revised.

Reviewer 3 Report

The manuscript Long-term immunogenicity of inactivated and oral polio vaccines: an Italian retrospective cohort study by Larocca AMV et al. is an interesting paper, which confirms high and lasting protective antibody response to both, inactivated and live polio virus vaccines. The paper is generally well written and the objective is clearly exposed. However, some minor points should be addressed.

In the Introduction the Authors rightly stress the difficulties in the polio eradication program set up by the WHO and the statement that the WHO European Region was declared to be polio free already in 2002.  Nonetheless, the WHO strategy of polio eradication may be slowed down in situations of conflict, in which socio-environmental and hygienic conditions may be disrupted. Emblematic is the situation in Ukraine, where a polio outbreak was registered from 2014 to 2017, following the Russian invasion of 2014 with special forces occupying part of the south-east area of Donbas and annexing the Crimea peninsula, thus creating a condition of local conflict (Quinn, V.J.M.; Dhabalia, T.J.; Roslycky, L.L.; Wilson, V.J.M.; Hansen, J.C.; Hulchiy, O.; Golubovskaya, O.; Buriachyk, M.; Vadim, K.; Zauralskyy, R.; et al. COVID-19 at War: The Joint Forces Operation in Ukraine. Disaster Med Public Health Prep. 2021 Mar 25:1-8. doi: 10.1017/dmp.2021.88).

In the Materials and Methods, the Authors should better specify when they collected the blood samples from the medical students and residents and how the anonymity of the samples was organized and assured. Moreover, the test for anti-polio 1, 2 and 3 neutralizing antibodies should be reported in extenso in a specific paragraph, as in the reference 12 cited by the Authors. Finally, the Statistical analysis should be reported in a dedicated paragraph.

In the Results, Table 1, total high titer anti-PV1, the percentage should be 91.4 instead of 91.5. Moreover, in the Table 1, which should be Table 2, with multivariate logistic regression analysis, chronic disease should be defined; which types of chronic disease, even in consideration that chronic diseases are not easily found in young people, and the maximum age was 33-year. Finally, the HCW mentioned in lines 175-176, are former medical students who have become HCW or do they refer to the residents cited in line 64? However, it should be useful to know how many medical students and how many residents, with their respective average age. Lastly, what are the hospitalizations for, cited in lines 175-176?

In the Discussion, the novelty of this paper is not clearly reported; in fact, the Authors report (lines 195-199) that this is the first study analyzing a large population vaccinated with four doses of OPV and comparing the response to OPV and IPV, whereas in the Introduction (lines 53-56) the persistence of neutralizing antibodies after OPV vaccination seemed a well-known data, whereas the persistence of neutralizing antibodies following IPV vaccination seemed unclear. Personally, I believe that the few studies of immunogenicity and persistence of vaccine-induced antibodies by oral and inactivated polio virus vaccines should be cited and compared with the current study, in order to let the reader better understand the novelty of this study.

Line 200: SIAs should be reported in extenso.

References

The reference 14 is incomplete.

Author Response

Q1. In the Introduction the Authors rightly stress the difficulties in the

polio eradication program set up by the WHO and the statement that the

WHO European Region was declared to be polio free already in 2002.

Nonetheless, the WHO strategy of polio eradication may be slowed down in

situations of conflict, in which socio-environmental and hygienic

conditions may be disrupted. Emblematic is the situation in Ukraine,

where a polio outbreak was registered from 2014 to 2017, following the

Russian invasion of 2014 with special forces occupying part of the

south-east area of Donbas and annexing the Crimea peninsula, thus

creating a condition of local conflict (Quinn, V.J.M.; Dhabalia, T.J.;

Roslycky, L.L.; Wilson, V.J.M.; Hansen, J.C.; Hulchiy, O.; Golubovskaya,

O.; Buriachyk, M.; Vadim, K.; Zauralskyy, R.; et al. COVID-19 at War:

The Joint Forces Operation in Ukraine. Disaster Med Public Health Prep.

2021 Mar 25:1-8. doi: 10.1017/dmp.2021.88).

A1. We added this study in introduction.

Q2. In the Materials and Methods, the Authors should better specify when
they collected the blood samples from the medical students and residents
and how the anonymity of the samples was organized and assured.
Moreover, the test for anti-polio 1, 2 and 3 neutralizing antibodies
should be reported in extenso in a specific paragraph, as in the
reference 12 cited by the Authors. Finally, the Statistical analysis
should be reported in a dedicated paragraph.

A2. Revised.

Q3. In the Results, Table 1, total high titer anti-PV1, the percentage
should be 91.4 instead of 91.5. Moreover, in the Table 1, which should
be Table 2, with multivariate logistic regression analysis, chronic
disease should be defined; which types of chronic disease, even in
consideration that chronic diseases are not easily found in young
people, and the maximum age was 33-year. Finally, the HCW mentioned in
lines 175-176, are former medical students who have become HCW or do
they refer to the residents cited in line 64? However, it should be
useful to know how many medical students and how many residents, with
their respective average age. Lastly, what are the hospitalizations for,
cited in lines 175-176?

A3. Revised.

Q4. In the Discussion, the novelty of this paper is not clearly reported; in
fact, the Authors report (lines 195-199) that this is the first study
analyzing a large population vaccinated with four doses of OPV and
comparing the response to OPV and IPV, whereas in the Introduction
(lines 53-56) the persistence of neutralizing antibodies after OPV
vaccination seemed a well-known data, whereas the persistence of
neutralizing antibodies following IPV vaccination seemed unclear.
Personally, I believe that the few studies of immunogenicity and
persistence of vaccine-induced antibodies by oral and inactivated polio
virus vaccines should be cited and compared with the current study, in
order to let the reader better understand the novelty of this study.

A4. We added few specific sentences in strength and limitations paragraph.

Q5. Line 200: SIAs should be reported in extenso.

A5. Revised.

Q6. References: The reference 14 is incomplete."

A6. Revised.